# Arising Novel Agents in Lung Cancer: Are Bispecifics and ADCs the New Paradigm?

**DOI:** 10.3390/cancers15123162

**Published:** 2023-06-13

**Authors:** Amanda Reyes, Rebecca Pharaon, Atish Mohanty, Erminia Massarelli

**Affiliations:** Department of Medical Oncology & Therapeutics Research, City of Hope National Medical Center, Duarte, CA 910102, USA; amareyes@coh.org (A.R.); rpharaon@coh.org (R.P.); amohanty@coh.org (A.M.)

**Keywords:** non-small cell lung cancer, EGFR mutated, immunotherapy, targeted therapy, bispecific antibodies, antibody-drug conjugates, EGFR resistance, MET amplification, Axl signaling, HER2, HER3, cMet signaling

## Abstract

**Simple Summary:**

Lung cancer is the leading cause of cancer-related death worldwide according to the World Health Organization. Non-small cell lung cancer makes up the majority of cases. Immunotherapy with immune checkpoint inhibitors and targeted therapy with tyrosine kinase inhibitors and other molecular targeted agents significantly changed the treatment landscape and overall survival. Unfortunately, resistance to these treatments develops, and there is a need to identify additional innovative therapies that can overcome treatment resistance. Advancements in biomedical engineering and technology allowed for the development of novel agents, capable of delivering effective treatment directly to tumor cells. These agents include antibody drug conjugates and bispecific antibodies which have various targets and mechanisms of action, discussed in depth in this review.

**Abstract:**

Lung cancer is one of the most common cancers with the highest mortality. Non-small cell lung cancer (NSCLC) contributes to around 85% of lung cancer diagnoses (vs. 15% for small cell lung cancer). The treatment of NSCLC has vastly changed in the last two decades since the development of immunotherapy and targeted therapy against driver mutations. As is the nature of malignancy, cancer cells have acquired resistance to these treatments prompting an investigation into novel treatments and new targets. Bispecific antibodies, capable of targeting multiple substrates at once, and antibody–drug conjugates that can preferentially deliver chemotherapy to tumor cells are examples of this innovation. From our initial evaluation, both treatment modalities appear promising.

## 1. Introduction

Lung cancer is the leading cause of cancer-related death worldwide according to the WHO [1]. Non-small cell lung cancer (NSCLC) contributes to around 85% of lung cancer diagnoses (vs. 15% for small cell lung cancer), which is further categorized into adenocarcinoma 78% and squamous 18% [2]. Since the early 2000s, the treatment options for NSCLC have exponentially increased after the discovery of immune checkpoint inhibitors and targetable ‘driver mutations’ [2,3]. Immune checkpoint inhibitors, pembrolizumab, atezolizumab, and cemiplimab were proven effective first-line agents in programmed cell death ligand 1 [PD-L1] positive metastatic NSCLC after the respective phase 3 trials [4,5,6]. Combination PD-L1 and cytotoxic T-lymphocyte-associated protein 4 [CTLA-4] inhibitors, nivolumab and ipilimumab were also proven in the first line compared to standard-of-care chemotherapy regardless of PD-L1 status [7]. 

The first successful targets of ‘driver mutations’ were epidermal growth factor receptor (EGFR)-activating mutations, most commonly deletions in exon 19 or point mutation L858R in exon 21, found in approximately 15% of adenocarcinomas [2,8,9,10]. Tumors with these mutations demonstrated significant sensitivity to tyrosine kinase inhibitors (TKIs; gefitinib and erlotinib) prompting investigation in several clinical trials [OPTIMAL, EURTAC, ENSURE, etc.]. These trials confirmed the superior ORR and median PFS of these first-generation TKIs versus standard-of-care chemotherapy and established their use as first-line therapy [11,12,13,14,15]. 

Genetic alterations developed, notably the T790M mutation, which conferred resistance to the EGFR first-generation agents [16]. From the landmark FLAURA trial, third-generation TKI osimertinib demonstrated effectiveness as first-line treatment with improved median PFS and OS compared to the first-generation agents, thereby becoming the new standard of care [17]. Additionally, numerous other driver mutations have entered the standard of care in first- and second-line treatment, notably ALK, ROS1, BRAF V600E, MET Ex14 skipping, RET, HER2, NTRK, and, most recently, KRAS G12C [18,19,20,21,22,23,24,25,26]. Unfortunately, resistance either through target-dependent (changes in the structure of the tyrosine kinase/target preventing inhibition) or target-independent (upregulation of bypass signaling pathways) emerged [27]. This resistance to treatment with molecular targeted inhibitors and immune checkpoint blockade prompted researchers to investigate mechanisms and pathways to circumvent it. This review will provide an analysis of some of these new treatments and pathway targets. 

## 2. Bispecific Antibodies

The concept of antibodies with the capability of targeting and utilizing multiple substrates (Bispecific, Trispecific) which could then be further enhanced to utilize T cells, so-called Bispecific T cell Engagers (BiTE), is relatively new in the context of cancer treatment. It was not until 1984 when BiTEs were first shown to employ T cells to prevent the growth of tumor cells in vivo [28]. Success in clinical trials was limited following the initial discovery, but molecular modifications discovered from preclinical data eventually translated into clinical results [29]. The first successful use of bispecific antibodies that did not utilize T cells was seen in end-stage Hodgkin’s lymphoma where one patient treated with an NK cell activating CD16/CD30 bispecific antibody achieved a complete remission while another achieved a partial remission from the study group of 15 patients [30]. Since then, advances in protein engineering and molecular techniques prompted a variety of formulations and targets but the most common remains bispecific antibodies that redirect effector immune cells [T cells] to target malignant cells [BiTE] [31]. The quick adoption of this treatment modality was not universal across the spectrum of malignancies, but interest has peaked recently, and lung cancer is no exception. 

### 2.1. C-Met Targeting

While the prevalence of EGFR resistance in NSCLC is increasing since the introduction and widespread use of TKIs, bispecific antibodies may provide a potential solution. EGFR mutations with insertions in exon 20, except A763_Y764insFQEA, have demonstrated very low response rates (<10%) to common EGFR TKIs (osimertinib, erlotinib, etc.), which is thought to be related to alterations that prevent TKI binding [32,33]. Additionally, MET gene amplification leads to the upregulation of cMet, and the subsequent pathway allows cancer cells to bypass EGFR signaling for cell survival, thereby creating another avenue of resistance to our current EGFR-targeted treatments [34]. The CHRYSALIS trial, a phase I trial dose-escalation/dose-expansion study, sought to focus on both resistance patterns with amivantamab, a dual EGFR and cMET bispecific antibody [35] (Figure 1). The initial data were promising with a PFS of 8.3 months [95% CI, 6.5 to 10.9] and an ORR of 40% [95% CI, 29 to 51] [34] (Table 1). The drug appears to have tolerable side effects which include infusion reactions most commonly seen during the first cycle, 49% grade 2/3 rash, and 20% grade 2/3 paronychia [35]. 

Additional investigation into amivantamab is currently underway including MARIPOSA1, a phase 3 trial evaluating the combination of amivantamab and lazertinib versus osimertinib in the first-line setting [36]. There are also early studies in the preclinical and phase 1 settings attempting to identify additional targets and combination agents involving cMet signaling. One such example is AZD-9592, a bispecific antibody that targets both EGFR and c-Met but is also conjugated to a topoisomerase inhibitor, which allows for both cytotoxic effects and receptor signaling blockade, currently recruiting in a phase 1 trial (NCT05647122) (Table 1). 

### 2.2. HER2/HER3 Targeting

After human epidermal growth factor 2 (HER2) targeting was utilized in breast cancer successfully, similar attempts were made in lung cancer. Bispecific antibodies were once again implemented to target multiple facets of cell regulation. Zenocutuzumab, a bispecific antibody designed with one Fab arm that binds to HER2, allows the other Fab arm targeting human epidermal growth factor 3 (HER3) to be in a position to prevent NRG1 binding to HER3, thereby blocking activation and subsequent cell signaling [37] (Figure 1). This drug was first proven in pre-clinical models of lung, breast, pancreas, and ovarian cancers with neuregulin-1 (NRG1) fusions in both in vitro and in vivo settings [38]. In the same study, a patient who had progressed on multiple lines of therapy with CD74-NRG1 NSCLC responded quickly with a partial response to treatment [38]. Currently, there is a phase II trial evaluating this drug in both NSCLC and pancreatic cancer in patients with NRG1 fusions (NCT02912949) (Table 1). Additional targeted therapy directed towards HER2/HER3 has entered the forefront of NSCLC treatment and will be discussed in later sections. 

### 2.3. PD-1/CTLA-4 Targeting

Immune checkpoint inhibitors have been at the center of NSCLC treatment for the last decade. Data from clinical trials in lung cancer, as well as melanoma and renal cell carcinoma, demonstrate that dual CTLA-4 and PD-L1 inhibition improves overall survival [7,39,40,41]. Unfortunately, there were increased rates of immune-related adverse events at the optimal effective treatment dose of CTLA-4/PD-L1 inhibitors [42]. This left a role for novel agents that are able to target both PD-L1 and CTLA-4 while maintaining tolerable toxicity. MEDI5752, a bispecific antibody of PD-1 and CTLA4, was specifically designed to bind CTLA-4- in PD-1-positive T cells allowing a higher level of T cell proliferation [43]. The Initial results of the phase 1b/II trial of MEDI5752 plus chemotherapy (carboplatin and pemetrexed) vs. pembrolizumab plus chemotherapy in the first line were presented at ESMO in September 2022; ORR was 50.0 [95% CI 27.2–72.8] with the study drug versus 47.6 [95% CI 25.7–70.2] in the pembrolizumab group [44]. Notable separation in ORR was seen in the PD-L1 < 1% group with ORR of 55.6 [95% CI 21.2–86.3] with the study drug and 30.0 [95% CI 6.7–65.2] (NCT03530397) (Table 1). Phase III trial data of this novel agent may provide additional insight into the efficacy and safety of this treatment modality. Another PD-L1 targeting agent, INBRX-105, a bispecific against PD-L1 and human 4-1BB receptor (CD137), is being investigated in a phase II study, used in conjunction with pembrolizumab in solid tumors including NSCLC (NCT03809624) (Table 1). CD137 has been shown to enhance T cell proliferation and T cell costimulatory functions and is therefore thought to act synergistically with ICI [45]. 

### 2.4. Small Cell Lung Cancer Targeting

Targeted therapy in small cell lung cancer (SCLC) has been disappointing in comparison to its NSCLC counterpart, but researchers have made substantial efforts to bridge this gap including an investigation into bispecific antibodies. In preclinical studies, DLL3 (delta-like ligand three) was found in high concentrations in SCLC tumor cells and was later found to be essential to tumorigenesis, prompting interest as a treatment target [46,47]. As rovalpituzumab tesirine was unsuccessful as an antibody–drug conjugate targeting DLL3 [48], researchers evaluated bispecific molecules that target DLL3, specifically, a bispecific T cell engager [BiTE] as an alternative (AMG757) [49]. Preclinical data on the novel agent AMG757 is promising with complete responses in both PDX models and orthotopic models due to successful T cell activation and subsequent T cell-induced tumor lysis [49]. Additionally, no treatment-related adverse events were reported with the drug [49]. In a phase I trial of this drug, now tarlatamab, 107 heavily pretreated patients with SCLC had an ORR of 23.4% [95% CI, 15.7 to 32.5] and median progression-free survival of 3.7 months [95% CI, 2.1 to 5.4] and 13.2 months [95% CI, 10.5 to not reached] [50]. Notable side effects include cytokine release syndrome and CNS toxicity [50]. This prompted a phase II trial which is currently ongoing (NCT05060016) (Table 1). 

## 3. Antibody-Drug Conjugates

To combat the toxicity of traditional chemotherapy agents, scientists started utilizing antibodies to preferentially deliver toxic chemotherapy directly to tumor cells, so-called antibody-drug conjugates (ADCs). ADCs are monoclonal antibodies targeted to a relevant tumor antigen and linked to a chemotherapy “payload” [51]. These agents were first utilized in human clinical trials in 1983 with the ADC, anti-carcinoembryonic antigen–antibody–vindesine conjugate [51]. However, the widespread success of these agents was not achieved until the early 21st century after FDA approval of gemtuzumab ozogamicin in 2000 for CD33-positive acute myeloid leukemia (later withdrawn in 2010, but reapproved by the FDA in 2017), brentuximab vedotin in 2011 for relapsed/refractory Hodgkin’s lymphoma, and tratuzumab emtansine in 2013 for HER2 positive breast cancer [52,53,54,55]. This success fueled investigation into other targets and malignancies, including lung cancer. 

### 3.1. HER2 Targeting

Due to the success of trastuzumab emtansine [TDM1] in HER2-positive breast cancer, there were similar hopes regarding its implementation into lung cancer, but this was more complicated with the variability in HER-2 evaluation. Currently, HER2 can be measured by protein expression [IHC], amplification [FISH cytology], and next-generation sequencing for mutations, but the results can be conflicting [56]. Initial studies of trastuzamab, a monoclonal antibody binding to HER2, alone in HER2 lung cancer (measured HER2 expression by IHC) were disappointing, but the combination with chemotherapy appeared more promising [57,58,59] (Figure 1). When trastuzumab was linked with ematansine, an antimicrotubule agent, forming an ADC, a phase II study (22 patients with HER2 exon 20 mutations) found an ORR of 38.1% [90% confidence interval, 23.0–55.9%] and median PFS and median OS of 2.8 and 8.1 months [59] (Table 2). An additional phase II trial of trastuzumab ematansine with several cohorts across different disease types that specifically analyzed NSCLC patients (N = 18 patients) with HER2 mutations by NGS rather than overexpression by IHC found ORR was 44% (95% CI, 22% to 69%) and median PFS was 5 months [95% CI, 3 to 9 months] [60] (NCT02675829). 

Additional formulations of ADCs with trastuzumab were created, most notably trastuzumab deruxtecan (topoisomerase I inhibitor) in which the payload is membrane-permeable, thereby facilitating cell death in nearby cells regardless of HER2 status, the so-called “cytotoxic bystander effect” [61]. This was translated into clinical practice in a phase I trial of 60 patients with HER2 expressing non-breast/GI or HER2 mutant solid tumors with an ORR of 28.3%, but, surprisingly, ORR was 72.7%, and median PFS was 11.3 [95% CI, 8.1–14.3] months in the HER2 mutant NSCLC patients [62]. Further, in DESTINY-Lung01, a phase II study of 91 patients with previously treated HER2-mutated NSCLC, ORR was 55% [95% CI 44–64], and PFS was 8.2 [95% CI 6.0–11.9] [50] (NCT03505710) (Table 2). Significant adverse events included interstitial lung disease which occurred in 26% of the patients and was fatal in two patients [63]. DESTINY-Lung04, a phase III trial of trastuzumab deruxtecan in the first line in unresectable or metastatic NSCLC with HER2 exon 19 or 20 mutations is ongoing (NCT0504879) and will potentially provide additional insight into this line of treatment. 

### 3.2. HER3 Targeting 

Along the same line as HER2-targeted therapy in NSCLC, investigators also sought to use ADCs in HER3 targeting. HER3 expression, while common across other solid tumors, is especially relevant in lung cancer as it is expressed in 83% of NSCLC with the highest expression in EGFR-mutated NSCLC and is correlated with progression and poor relapse-free survival [64,65]. Deruxtecan was once again utilized as the drug conjugate in combination with a HER3 antibody in the novel agent, patritumab deruxtecan [66] (Figure 1). Initial data in a phase I trial of 57 patients revealed an ORR of 39% [95% CI, 26.0–52.4], and a median PFS of 8.2 [95% CI, 4.4–8.3] months [66]. The HERTHENA-Lung02 trial, a phase 3 trial of patritumab deruxtecan vs. platinum-based chemotherapy in EGFR-mutated NSCLC after EGFR TKI failure is enrolling at present (NCT05338970) (Table 2). From analysis of data from previous trials and future studies of HER3, we may be able to identify a biomarker of treatment responsiveness in addition to just HER3 expression which could help guide clinicians in the future. 

### 3.3. TROP2 Targeting 

As modifications and formulations of HER2/HER3 targeting are ongoing, new targets are also examined in this clinical space. Like HER, trophoblast cell surface antigen 2 (TROP-2), a transmembrane glycoprotein, is common among epithelial cancers including NSCLC [67]. Additionally, a large meta-analysis of 16 studies with 2569 patients found the overexpression of TROP-2 was associated with poor prognosis including overall survival with HR 1.896 [95% CI 1.599–2.247] and disease-free survival (DFS) 2.336 [95% CI 1.596–3.419] [68]. Therefore, TROP-2 was evaluated in clinical trials; the first successful use was a TROP-2 antibody conjugated to topoisomerase I inhibitor, sacituzumab govitecan, in metastatic triple-negative breast cancer with significantly improved PFS and OS compared to single-agent chemotherapy (physician choice of eribulin, vinorelbine, capecitabine, or gemcitabine) in the phase III ASCENT trial [69].

In pre-clinical studies of lung cancer, high TROP-2 expression was associated with lymph node metastasis and high histological grade; in cell lines, its overexpression increased cell proliferation and invasion [70]. This prompted further evaluation into TROP-2-based therapy in the lung along with other solid tumors with TROPION-PanTumor01 [71]. In a phase I trial of ADC, datopotamab deruxtecan (which targets TROP2) found ORR using a dose of 8 mg/kg, 25% (20/80); 6 mg/kg, 21% (8/39); and 4 mg/kg, 23% (9/40) with increased treatment-associated adverse events in the 8 mg dose group, including pneumonitis at 15% compared to 2% at the lower doses [71] [Figure 1]. TROPION-Lung01, a phase III trial to evaluate datopotamab deruxtecan vs. docetaxel in patients with advanced/metastatic NSCLC without an actionable mutation is ongoing (NCT04656652) (Table 2). Other investigations into this drug seek to evaluate its effectiveness in the NSCLC population with actionable mutations after receiving platinum-based chemotherapy with a phase 2 trial TROPION-Lung05 (NCT04484142). These ongoing trials will hopefully answer some of the questions regarding TROP-2 targeting and determine where it will fall in the hierarchy of treatment. 

### 3.4. C-Met Targeting 

As there are limited treatment options after EGFR TKI resistance in the EGFR-mutated NSCLC population, there has been a focus on the development of therapies aimed at this population. The trial of Teliso V, an ADC of a c-met antibody and monomethyl auristatin E (microtubule inhibitor), investigated the effectiveness of the drug in several different populations of NSCLC patients to determine which cohort has the most optimal response [72] (Figure 1). Initial results presented at ASCO 2022 reported an ORR of 36.5% in the non-squamous EGFR wild-type cohort but notably 52.2% ORR in the c-Met high group and 24.1% in the c-met intermediate group [73]. Surprising, both the non-squamous EGFR-mutant NSCLC group and the squamous NSCLC had a less robust response regardless of c-Met expression with ORR ranging from 11.1% to 16.7%, which prompted the closure of these cohorts/cessation of enrollment per study protocol [73]. Teliso V does appear to be well tolerated, with the most common adverse side effects being peripheral neuropathy (25%), nausea (22%), and hypoalbuminemia (20.6%) [73]. The phase 2 trial in patients with previously treated c-Met+ NSCLC is in progress (NCT03539536) (Table 2). The combination with EGFR TKIs has been promising for combating EGFR resistance; a phase 1/1b study of osimertinib therapy in combination with Teliso-V showed ORR of 50% in heavily treated, c-MET-high patients after the failure of Osimertinib monotherapy [74]. In addition, a study of erlotinib plus Teliso-V therapy had a PFS of 5.9 months, an ORR of 32.1%, and a DCR of 85.7% in patients previously treated with EGFR TKIs including c-MET-high-expressed patients [75].

### 3.5. AXL Targeting 

In preclinical evaluations, Axl, a receptor tyrosine kinase that is part of the TAM (TYRO3, AXL, and MER) family, was shown to have several roles in cancer development including cell growth, adhesion, survival, and proliferation [76,77]. In addition, AXL is associated with higher-grade cancers, treatment resistance, and overall poor prognosis [76,78]. It is of no surprise that it has been a treatment target in many solid and hematologic malignancies [76]. In NSCLC, the activation of AXL kinase induced resistance to EGFR TKIs (erlotinib in this study), and inhibiting AXL functioning restored TKI sensitivity [79]. Enapotamab vedotin is an AXL-specific ADC that demonstrated proven activity in NSCLC in the preclinical setting [80]. The drug was evaluated across multiple tumor types in a phase I trial, but the clinical response was disappointing; therefore, the development of the drug was halted [81,82] (Table 2). Additionally, a novel ADC targeting AXL with a pyrrolobenzodiazepine cytotoxin, called Mipasetamab uzoptirine, showed promise in preclinical evaluation and is currently in a phase I clinical trial (NCT05389462) [83] (Table 2). 

### 3.6. Emerging ADCs

After the success in the abovementioned ADCs, researchers are evaluating other combinations and molecules that may have similar or improved efficacy. Nectin-4 (cell adhesion molecule), tissue factor, and CEACAM5 (carcinoembryonic antigen-related cell adhesion molecule 5) were investigated in phase I trials in solid tumors with promising results and are now in phase II/III trials [84,85,86] (Table 2). Other molecules have been less successful in NSCLC, such as mesothelin, NaPi2b (sodium-dependent phosphate transporter), PTK7 (protein tyrosine kinase 7), and folate receptor alpha, and were not advanced past phase I [87,88,89,90] (Table 2).

## 4. Conclusions

Over the last two decades, the landscape of NSCLC treatment has drastically changed from the days of platinum doublet chemotherapy. Treatment options are now numerous with a significant proportion of targeted agents (NCCN NSCLC guidelines). However, responses to treatment are not forever. As is the nature of malignancy, cancer cells often develop mechanisms and alternative pathways to circumvent treatment-induced cell death. The development of ‘multi-drug resistance’ [MDR] is one postulated mechanism by which metastatic disease becomes refractory to treatment [91]. Cancer cells can develop resistance through a multitude of processes including repairing DNA damage, preventing treatment-induced apoptosis, and many others [92]. Even with treatments that target a known resistance pattern, an additional genetic mutation can render the treatment obsolete or ineffective, notably demonstrated by the acquired EGFR C797S mutation in NSCLC with preexisting EGFR T790M mutation [93]. For these reasons, investigators had to design innovative drugs that could overcome these problems.

The idea of preferentially delivering a chemotherapy agent to a cancerous cell via an antibody, antibody–drug conjugates, was one such innovative design. Another aspect of scientific breakthroughs are bispecific antibodies with the thought that targeting multiple tumor markers at the same time or bringing cytotoxic T cells in close proximity to the tumor cells could overcome known resistance. On the cumulative evaluation of these ADCs and bispecific antibodies, it does appear that they are more effective when used in a ‘precision medicine’-type approach focusing on a specific population most likely to benefit (with a specific mutation, biomarker, expression) rather than the generalized all patients/all tumor types approach utilized in typical chemotherapy. On the other hand, the pre-screening requirements of tissue biopsies (if even possible) and molecular testing substantially delay treatment initiation, so a universal agent with efficacy in all NSCLC patients could be beneficial.

One potential concern of these new agents is longevity and long-term side effects. Given the somewhat recent discovery and implementation, there is no significant follow-up data. Additionally, sequencing the novel agents with our current FDA-approved agents may lead to unforeseen complications, as was seen with the higher rates of pneumonitis with EGFR TKIs after immune checkpoint inhibitors [94]. Moreover, these ADCs/bispecifics are attempting to replace or at least be utilized as long-term maintenance therapy, like TKIs, but this is potentially problematic as the ‘payload’ (conjugated drug) is a chemotherapy agent, which are traditionally used in shorter intervals. The toxicity of the payload is influenced by several factors, including the drug–antibody ratio, the type of linker (cleavable vs. non-cleavable), and if the target is expressed commonly on adjacent normal tissue, among others [95,96].

Treatment penetration into the blood–brain barrier varies substantially by drug and has been an important factor in the disease control of NSCLC. We lack substantial data on these new drugs to determine the blood–brain barrier permeability, but a high concentration in the CSF might not be required. In prior studies, intravenous pembrolizumab demonstrated efficacy in the brain despite low levels in the CSF [97,98,99]. There is still much to discover about bispecific antibodies and antibody–drug conjugates in the treatment of lung cancer, but we can conclude that these treatment modalities are here to stay.

## Figures and Tables

**Figure 1 cancers-15-03162-f001:**
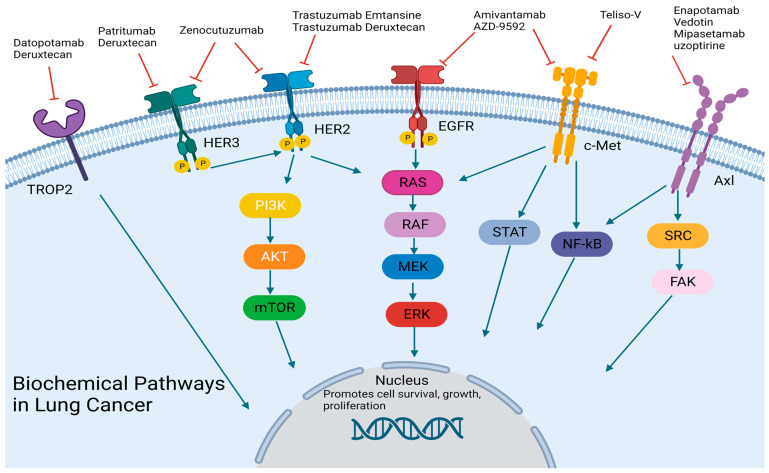
Biochemical Pathways: antibody-drug conjugates and bispecific antibodies targets Designed using BioReader.

**Table 1 cancers-15-03162-t001:** Bispecific antibodies in lung cancer.

Drug Name	Trial	NCT
Amivantamab	MARIPOSA1/Phase 3	NCT04487080
AZD-9592	EGRET/Phase 1	NCT05647122
Tarlatamab	DeLLphi-301/Phase 2	NCT05060016
MEDI5752	Phase 1	NCT03530397
Zenocutuzumab	Phases 1 and 2	NCT02912949
INBRX-105	Phase 2	NCT03809624

Trial data obtained from NIH clinicaltrials.gov.

**Table 2 cancers-15-03162-t002:** Antibody-Drug Conjugates in Lung Cancer.

Drug Name	Trial	Target	NCT
Enapotamab Vedotin	Phase 1	Axl	NCT02988817
Trastuzumab Emtansine	Phase 2	HER2	NCT02289833
Trastuzumab Deruxtecan	DESTINY-Lung01	HER2	NCT03505710
Patritumab Deruxtecan	HERTHENA-Lung02	HER3	NCT05338970
Datopotamab Deruxtecan	TROPION-Lung01	TROP2	NCT04656652
Teliso-V	Phase 2	c-Met	NCT03539536
Mipasetamab uzoptirine	Phase 1	Axl	NCT05389462
Enfortumab Vedotin	Phase 2	Nectin-4	NCT04225117
Tisotumab Vedotin	InnovaTV 201	Tissue Factor	NCT02001623
Farletuzumab Ecteribulin	Phase 1	Folate receptor alpha	NCT03386942
Cofetuzumab Pelidotin	Phase 1	PTK7	NCT02222922
Tusamitamab Ravtansine	CARMEN-LC03	CEACAM5	NCT04154956
Lifastuzumab Vedotin	Phase 1	NaPi2b	NCT01363947

Trial data obtained from NIH clinicaltrials.gov.

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
