# Peer review of "Arising Novel Agents in Lung Cancer: Are Bispecifics and ADCs the New Paradigm?"

_cancers, 2023, doi:10.3390/cancers15123162_

Round 1

Reviewer 1 Report

Overall well written and comprehensive review of a timely and important topic.

A few comments:

In Section 1, page 2, line 60 - for that paragraph would more clearly distinguish bispecific antibodies from BITE's.  You start by discussing bispecific antibodies but then transition to BITE's without any clarification of the difference between the two.

Figure 1 on page 2 - the resolution seem poor

On page 3, line 93, would add that amivantamab can cause infusion reactions esp with the first cycle as well as 49% grade 2/3 rash, 20% grade 2/3 paronychia

On page 4, line 135, might omit "in widespread clinical practice" as it does not add much

On page 4, line 137, might add brief sentence about rationale for CD137 as a target in NSCLC

On page 4, line 145, I ma not sure what "this" is referring to but I assume it is rovalpituzumab tesirine - might include a sentence clarifying what approach was "unsuccessful" and why

On page 4, line 154, might add brief statement about toxicity with tarlatamab, namely CRS and CNS effects

On the top of page 5, there may need to be a new heading to clarify that this is a new topic

On page 5, line 159 might further clarify/define ADC's - with something like (just a suggestion) "ADC's are monoclonal antibodies to a relevant tumor antigen that are linked to a chemotherapy "payload" - perhaps after the first sentence.

On page 6 in the HER-2 targeting paragraph, would suggest adding a statement explaing the various ways her-2 is altered/evaluated in tumors, i.e. protein expression by IHC, amplification by FISH, and her-2 mutations, as the trials you discuss touch on all of these different populations

On page 6, line 206, would add "NSCLC" after "mutated"

Page 6, lines 217-19, this sentence is no clear and vague as written.  Could be re-written or omitted.

On page 7, line 227, the sentence starting "Therefore" is not clear as written

Page 8, lines 280-84, the paragraph could be omitted as it is not really relevant to the topic of the paper.  if you keep it, would move "NSCLC" in line 280 to after the word "mutant"

on page 8, line 286, the phrase "As success in the above-mentioned ADC's" does not make sense as written

In addition, on page 8, lines 293-95 the discussion on mesothelin in mesothelioma is not really germane to the topic and could be omitted.

overall I think the conclusion could be more concise and tightly written. The discussion on IHC and AI (lines 324-27)  is not really relevant as few of the drugs discussed rely on IHC.  In addition, the concluding discussion on CNS penetration is quite vague and speculative, including comparing the possible toxicity to that seen with CAR T-cell therapy.  It would be fine to acknowledge that failure in the brain is a common issue in NSCLC and therefore that CNS penetration of these agents will be clinically relevant, but I might leave it at that. 

Some other specific comments - page 8, line 308, need to add "with" prior to "treatments"

page 8, line 319, need to add "a" before "precision oncology" and "on" before "a specific"

Lines 328 and 329, would start a new sentence after "effects"

Line 342 and 343, the sentence is not clear as written

The English overall is fine, though I have made some suggested edits above.

Reviewer 2 Report

This manuscript is a review of current bispecifics and ADCs for lung cancer, and this is comprehensive and very well written. I offer some suggestions for revision to make the review even better.

1.     L46 of p.2, the reference.

According to the context, all comparison studied may as well be referred, then the two pivotal studies were excluded. I recommend that this two studies as referrence as below: (1)Maemondo M, et al. N Engl J Med 2010;362:2380-8. (2) Tony MK, et al. N Engl J Med 2009;361:947-57.

2.     L53 of p.2, ALK,ROS1,MET,RET,HER2,NTRK and most recently KRAS G12C

The author should add BRAF to this list. In addition, BRAF and MET should be changed to "BRAF V600E" and "MET Ex14skipping" according to the KRAS G12C spelling.

3.     L56 of p.2, the term of targeted tyrosine kinase inhibitors [TKIs]

This term can probably be changed to "(molecular) targeted inhibitors", "(molecular) targeted therapy". Because KRAS inhibitors or BRAF inhibitors are not tyrosine kinase inhibitors.

4.     L82-83, p3, the description about Ex20ins

I realize this is very trivial and not worth worrying about, but I am wondering if I should add the phrase "other than FQEA" since several data show that FQEA is strictly speaking also susceptible to first-generation EGFR TKIs. I would like to leave the decision to the author or editors.

5.     L118 of p.3, the number of section

Please check the section number. It should probably be changed to "1.3".

6.     L131 of p.4, the reference of ESMO

For clarifying the data of reference, I recommend adding the following citation as a reference to the study of MEDI5752 for NSCLC: Annals of Oncology (2022) 33 (suppl_7): S808-S869.

7.     L158 of p.4, the section number

The correct number is probably 2.

8.     L210 of p.6, the term of RECIST

Why was only this ORR accompanied by the term "RECIST v1.1"? The response rates that have been reported recently have all been evaluated by RECIST v1.1. I think that this term can be omitted.

9.     L260 of p.7, about the information of Teliso-V for EGFR-mutated population.

The authors should mention the study of EGFR TKI in combination with Teliso-V for this population to avoid misunderstanding. Phase 1/1b study of osimertinib therapy in combination with Teliso-V showed ORR of 50% even in heavily treated, c-MET high expressed patients after failure of osimertinib monotherapy. (J Clin Oncol 40, 2022 (suppl 16; abstr 9013)) In addition, erlotinib plus Teliso-V therapy  also showed clinical benefit, PFS of 5.9 months, ORR of 32.1%, DCR of 85.7%, in the population  including the c-MET high expressed patients pretreated with EGFR-TKI. (Journal of Clinical Oncology 41, no. 5 (February 10, 2023) 1105-1115.) I recognize that these data indicated the potential that Teliso-V can be therapeutic agent for resistant EGFR-mutated, c-MET expressed NSCLC under combination with EGFR-TKI.

10.  L 346 of p.9, CART

Is it correct that this abbreviation stands for "amphetamine-regulated transcript peptide"? Please describe in detail what it stands for.

Round 2

Reviewer 2 Report

The areas I pointed out have been corrected with almost no problems. One point needs to be corrected regarding FQEA, perhaps due to a misunderstanding.

There are several variants of Ex20ins, all of which are known to be less effective than conventional EGFR TK. However, among the Ex20ins, FQEA has been reported to be effective with erlotinib (JTO Clin Res Rep. 2020 Sep;1(3):100051), and there have been recent reports in Nature that afatinib may be effective for FQEA due to its structure (Nature. 2021 Sep;1(3):100051).  The authors state that "EGFR mutations with insertions in exon 20, A763_Y764insFQEA, have shown very low response rates (<10%) to common EGFR TKIs (osimertinib, erlotinib, etc.), which is thought to be related to alterations that prevent TKI binding. Consider changing the statement to "except FQEA".

Author Response

We had intended to include except prior to FQEA addition. It was accidently omitted. It has been replaced.